# Sputum Microbiota in Coal Workers Diagnosed with Pneumoconiosis as Revealed by 16S rRNA Gene Sequencing

**DOI:** 10.3390/life12060830

**Published:** 2022-06-02

**Authors:** Vladimir G. Druzhinin, Elizaveta D. Baranova, Ludmila V. Matskova, Pavel S. Demenkov, Valentin P. Volobaev, Varvara I. Minina, Alexey V. Larionov, Snezana A. Paradnikova

**Affiliations:** 1Department of Genetics and Fundamental Medicine, Kemerovo State University, 650000 Kemerovo, Russia; laveivana@mail.ru (E.D.B.); vminina@mail.ru (V.I.M.); alekseylarionov09@gmail.com (A.V.L.); bird.doctor@inbox.ru (S.A.P.); 2Institute of Living Systems, Immanuel Kant Baltic Federal University, 236016 Kaliningrad, Russia; liudmila.matskova@ki.se; 3Department of Microbiology, Tumor Biology and Cell Biology (MTC), Karolinska Institutet, 171 65 Stockholm, Sweden; 4Institute of Cytology and Genetics SB RAS, 630090 Novosibirsk, Russia; demps@bionet.nsc.ru; 5Center for Genetics and Life Sciences, Sirius University of Science and Technology, 354340 Sochi, Russia; volobaev.vp@gmail.com

**Keywords:** coal worker’s pneumoconiosis, sputum microbiome, lung fibrosis, next-generation sequencing, 16S rRNA genes, *Streptococcus*

## Abstract

Coal worker’s pneumoconiosis (CWP) is an occupationally induced progressive fibrotic lung disease. This irreversible but preventable disease currently affects millions across the world, mainly in countries with developed coal mining industries. Here, we report a pilot study that explores the sputum microbiome as a potential non-invasive bacterial biomarker of CWP status. Sputum samples were collected from 35 former and active coal miners diagnosed with CWP and 35 healthy controls. Sequencing of bacterial 16S rRNA genes was used to study the taxonomic composition of the respiratory microbiome. There was no difference in alpha diversity between CWP and controls. The structure of bacterial communities in sputum samples (β diversity) differed significantly between cases and controls (pseudo-F = 3.61; *p* = 0.004). A significant increase in the abundance of *Streptococcus* (25.12 ± 11.37 vs. 16.85 ± 11.35%; *p* = 0.0003) was detected in samples from CWP subjects as compared to controls. The increased representation of *Streptococcus* in sputum from CWP patients was associated only with the presence of occupational pulmonary fibrosis, but did not depend on age, and did not differ between former and current miners. The study shows, for the first time, that the sputum microbiota of CWP subjects differs from that of controls. The results of our present exploratory study warrant further investigations on a larger cohort.

## 1. Introduction

Coal worker’s pneumoconiosis (CWP) is an occupationally induced progressive fibrotic lung disease, caused by the deposition of coal mine dust in the lung parenchyma and by the subsequent tissue reaction. This public health problem typically occurs in the coal mining industry, including opencast mining, around the world [1,2]. In many countries, coal is still used as an important source of energy, and coal mining remains a major industry. This irreversible but preventable disease currently affects millions across the world [3]. Numerous studies have shown that inhalation of coal dust containing crystalline silica (silicon dioxide), usually in the form of quartz or silica, is the primary cause of silicosis, leading to progressive pulmonary fibrosis, which is the main clinical and pathological feature of CWP [4]. CWP is associated with an increased risk of malignancy [5], which is consistent with an increase in the baseline level of chromosomal damage in lymphocytes of patients compared with healthy individuals [6,7].

The main mechanisms of CWP include silica-induced macrophage cytotoxicity, activation of leukocytes to produce active oxygen radicals, and damage to alveolar epithelial cells stimulating fibroblast proliferation. Deregulation of DNA methylation was also pointed out as a possible mechanism of CWP pathogenesis [8]. However, the exact mechanisms of progressive pulmonary fibrosis in CWP remain to be elucidated. In particular, the possible effects of the respiratory tract microbiota on the etiology and pathogenesis of CWP need further investigation.

Numerous recent studies using metagenomic sequencing have shown that the respiratory microbiota plays an important role in maintaining lung health but can differ significantly in various diseases associated with the lungs [9,10]. Changes in the taxonomic composition of respiratory microbiota have been evaluated in patients with various pulmonary disorders: COPD [11], asthma [12], community-acquired pneumonia [13], cystic fibrosis [14], lung cancer [15], and idiopathic pulmonary fibrosis [16].

In the current study, the association between CWP and the taxonomic composition of respiratory microbiota was investigated. These changes may, in turn, be associated with progressive pulmonary fibrosis, which is regarded as the main clinical and pathological feature of CWP. Recently, several reports have highlighted the role of the microbiota in fibrosis affecting several human organs, i.e., intestine [17], cardiac tissue [18], liver [19,20], skin [21], and breast tissue [22,23]. Thus, data on the changes in the microbiome composition in the respiratory tract during CWP pathogenesis may be of importance to clarifying the role of microbiota in lung fibrosis.

To test this hypothesis, an initial analysis of the taxonomic composition of the sputum microbiome of coal miners suffering from CWP and from healthy subjects was performed using 16S ribosomal RNA sequencing.

## 2. Methods

### 2.1. Cohort Information

The composition of the bacterial microbiome in sputum samples was studied in 35 patients with CWP diagnosis (men only, average age 58.5 ± 8.3years) who were admitted to the Department of Occupational Disease Pathology, Kemerovo Regional Clinical Hospital (Kemerovo, Russian Federation). The diagnosis of CWP (code J60 according to ICD-10) was made on the basis of chest x-ray and spirography. All patients worked as underground coal miners. Of these, 8 (22.9%) were actively working in coal mines at the time of the survey, and 27 participants (77.1%) had ceased working due to the onset of the disease. Mining work experience in CWP patients varied from 18 to 37 years (average value 26.9 ± 5.3 years). As a control group, we examined 35 healthy male donors at a blood transfusion station, who were residents of Kemerovo (average age 55.7 ± 11.72 years). There were no significant age differences between CWP and controls. Among CWP patients there was one active smoker and among the controls all were non-smokers. The summarized information on CWP patients and controls is shown in Table 1. An individual questionnaire was filled out for each survey participant, containing information about the place and date of birth, profession, exposure to occupational hazards, health status, diet features, medications, X-ray records, and harmful habits (smoking and alcohol use). In the study, inclusion criteria were male ≥40 years of age and exclusive criteria any acute or chronic condition that would limit the ability of the patient to participate in the study, use of antibiotics within 4 weeks prior to collection, failure to obtain a sputum sample.

### 2.2. Sample Collection, Processing and Storage

To analyze the composition of the respiratory tract microbiome, samples of 2–3 mL sputum from CWP patients and controls were obtained over a period of 15–30 min, prior to all diagnostic or therapeutic procedures. Sputum samples were collected non-invasively through participant-induced coughing, (i.e., without induction) and represented the oropharyngeal secretion. Before sputum collection, all patients and controls were asked to rinse their mouths. Microscopic examination of Giemsa-stained cytological slides was used to and confirm the presence of columnar airway epithelial cells in random sputum samples. The resulting samples were immediately placed in sterile plastic vials and frozen (−20 °C). Frozen samples were transported to the laboratory and stored at −80 °C.

### 2.3. DNA Extraction, 16S rRNA Amplification and 16S rRNA Sequencing

Procaryotic DNA samples were extracted using FastDNA Spin Kit For Soil (MP Biomedicals) based on the manufacturer’s recommendation. From each sample, 500 µL of sputum was used for DNA extraction. The DNA concentration was monitored using the Qubit^®^ dsDNA Assay Kit and the Qubit^®^ Fluorometer (Life Technologies, CA, USA). Fifty nanograms of each of the extracted and purified sputum DNAs were used for the subsequent amplification of 16S rRNA genes. Amplification of 16S rRNA was carried out according to the Illumina protocol «Preparing 16S Ribosomal RNA Gene Amplicons for the Illumina MiSeq System». The approximately 500 bp long 16S rRNA amplicons consisted of a fragment within the hypervariable (V3–V4) region of the bacterial 16S rRNA genes. The initial PCR was performed with broad-spectrum 16S rRNA primers. In the next round of PCR, with the index-containing primers, fragments of approximately 630 bp length were produced. The Illumina 600 cycle MiSeq Reagent Kit V3 was used. As negative control, a contamination control sample without biological material was included at the sample preparation stage. DNA extraction was then performed in parallel from all samples. Next, the contamination control underwent 16S amplification and was visualized on a gel. As a positive control, a ZYMO community control probe was used (ZymoBIOMICS™ Microbial Community Standards, Cat No D6300 and D6310, Tustin, CA, USA).

The following 16S rRNA primers were used: forward primer: 5′- TCGTCGGCAGCGTCAGATGTGTATAAGAGACAGCCTACGGGNGGCWGCAG-3′. Reverse primer: 5′-GTCTCGTGGGCTCGGAGATGTGTATAAGAGACAGGACTACHVGGGTATCTAATCC-3′

Amplification was performed using BioMaster Hi-Fi LR 2× ReadyMix DNA polymerase (BiolabMix company, Novosibirsk, Russia). The primer sequence was taken from the recommended library preparation protocol for sequencing on the MiSeq platform https://support.illumina.com/documents/documentation/chemistry_documentation/16s/16s-metagenomic-library-prep-guide-15044223-b.pdf (accessed on 27 November 2013).

Cycle conditions were 94 °C (3 min 30 s), followed by 25 cycles of 94 °C (30 s), 55 °C (30 s), 68 °C (40 s), and a final extension at 68 °C (5 min). Libraries (~550 bp) were purified using Agencourt AMPure XP beads (Beckman Coulter, Brea, CA, USA). Dual indices and Illumina sequencing adapters from the Illumina Nextera XT index kits v2 B and C (Illumina, San Diego, CA, USA) were added to the target amplicons in a second PCR step using BioMaster Hi-Fi LR 2× ReadyMix DNA polymerase (BiolabMix company, Novosibirsk, Russia). Cycle conditions were 94 °C (3 min 30 s), then 8 cycles of 94 °C (30 s), 55 °C (30 s), 68 °C (40 s), and a final extension of 68 °C (5 min). Libraries (~630 bp) were again purified using XP beads. Preparation of 16S rRNA libraries was completed according to the Illumina 16S metagenomic sequencing library protocol. Sample PCR products were pooled in equimolar ratio, purified using XP Beads, and quantified using a fluorometer (Quantus Fluorometer dsDNA, Promega, Madison, WI, USA). Molarity of the libraries was brought to 4 nM, and the libraries were denatured and diluted to a final concentration of 8 pM with a 10% PhiX-DNA spike to align sequencing on the Illumina MiSeq (MiSeq Reagent Kit V3, 600 cycles) [24].

### 2.4. Taxonomy Quantification Using 16S rRNA Gene Sequences and Statistical Methods

The resulting data were processed using the program QIIME2 [25]. A quality check was carried out with the DADA2 software, the default parameters were used [26] and a sequence library was generated.

Amplicon sequence variants (ASV) sequences were performed with QIIME2, using naive Bayesian classifier models based on a 99% nucleotide composition similarity threshold using the Greengenes (versions 13-8) and SILVA (version 132) reference sequence library, followed by removing singletons (ASVs containing only one sequence).

The total diversity of prokaryotic sputum communities (alpha diversity) is estimated by the number of allocated ASVs (analog of species richness) and Shannon indices (H = Σp_i_ ln p_i_, p_i_–part of *i*-sh species in community) [27]. When calculating sample diversity indices, 498 sequences were normalized (the minimum number of received sequences per sample). The variation in the structure of the bacterial community of different samples (beta diversity) was also analyzed using UniFrac [27]—a method common in microbial ecology that estimates the difference between communities based on the phylogenetic relationships of the presented taxa. We used a version of the weighted UniFrac method. The significance of differences between groups of samples was evaluated by the PERMANOVA method (Adonis). Principal Coordinate Analysis (PCOA) graph construction was carried out by using QIIME2 package.

In addition, to assess the significance of differences in the relative percentage of individual bacterial taxa in the sputum samples, the Mann–Whitney U test was used. Spearman’s correlation coefficient was used to calculate correlations [28]. Calculations were performed using the software package STATISTICA.10, Statsoft, Tulsa, OK, USA. The false discovery rate (FDR) correction was used to assess the significance of differences in the relative percentages of individual bacterial taxa taking into account multiple comparisons. Multiple linear regression (MLR) was performed to predict the relationship between the relative abundance of individual bacteria in the sputum of CWP patients and lifestyle/disease factors. Additionally, linear discriminant analysis (LDA), effect size (LEfSe), as well as ANCOMBC analyses, were used to normalize data on observed microbial abundance.

Sequence data has been submitted and is archived in a public NCBI database. Archive number PRJNA820569.

## 3. Results

In the current study using 16S rRNA sequencing (V3–V4) in CWP patients and control subjects using sputum samples, a total of eight phyla with relative frequencies above 0.1% were identified. The prevailing phyla in our dataset were Firmicutes, Bacteroidetes, Actinobacteria, and Proteobacteria (Figure 1), as expected from previous studies [29,30]. For these eight types of bacteria, there were no differences between patients and controls (Table 2). ASVs in CWP subjects amounted to 142.0 ± 56.04, while in the control group–147.43 ± 43.6 (*p* > 0.05). Regarding alpha diversity, neither the number of allocated ASVs nor the Shannon indices showed any significant differences between CWP and controls. Overall, bacterial communities in the two groups in the study were fairly diverse as indicated by the Shannon index at the genus level (6.466 in CWP vs. 6.545 in controls).

Differences in the structure of bacterial communities in sputum samples of CWP and controls are shown in Figure 2. The PERMANOVA (Adonis) test using the distance matrix constructed by the weighted UniFrac method showed a significant difference in the prokaryotic communities in the sputum of healthy subjects as compared to miners with CWP patients (pseudo-F = 3.61; *p* = 0.004). Sequencing statistics for 22 genera (with no less than 0.1% relative frequency) are summarised in Table 3, alongside the corresponding U-rank Mann–Whitney *p* values and also taking into account the FDR amendment. Among genera, *Streptococcus*, *Prevotella* (*f. Prevotellaceae*), *Veillonella,* and *Anaerosinus* were the most common in the two pools.

In the sputum of CWP subjects, compared to controls, there was an increase in abundance (by percentage) of the following genera: *Streptococcus* (25.12 ± 11.37 vs. 16.85 ± 11.35; *p* = 0.0003); *Granulicatella* (1.71 ± 1.68 vs. 1.01 ± 1.39; *p* = 0.03) and *Bacillus* (2.67 ± 2.19 vs. 1.67 ± 1.7; *p* = 0.04). At the same time, the genus *Selenomonas* was less represented in the microbiomes of CWP subjects compared to the controls (2.07 ± 2.1 vs. 4.6 ± 4.39; *p* = 0.02). It should be noted that taking into account the FDR correction for multiple comparisons, only differences in *Streptococcus* abundance reached the confidence threshold.

In addition, we applied methods of discriminant analysis to find the most significant characteristics that distinguish groups of data from each other. The LEfSe method revealed an increase in the abundance of a number of taxa both in the CWP group (red) and in the control group (green) (Figure 3). The ANCOMBC method revealed an increase in the absolute abundance, in the CWP group relative to the control, for *Streptococcus* (beta = 1.86, W = 3.94, SE = 0.47, q-value = 0.01).

Sequencing statistics for 31 species (which met with a relative frequency of not less than 0.1%) are summarized in Table 4, alongside the corresponding U-rank Mann–Whitney *p* values and also taking into account the FDR amendment. The only significant difference between the sputum microbiome of CWP patients and controls may be the presence of the most common species, *Streptococcus agalactiae*. The average percentage of *Streptococcus agalactiae* in CWP sputum samples was significantly higher than in controls (25.32 ± 11.55 vs. 16.93 ± 10.92; *p* = 0.0002).

Given the fact that age can influence the composition of the microbiome, Spearman’s coefficient to assess the effect of age on the representation of all bacterial taxa was used. In the total sample collection (CWP patients and control subjects), an age-correlated increase in the abundance of the genus *Streptococcus* was observed; however, the significance of this increase was found only at the confidence limit (r = 0.217; *p* = 0.076). The correlation between age and the representation of other genera in the sputum samples was also insignificant.

To assess the possible correlations of disease duration to the content of different bacteria in sputum, Spearman’s coefficient was used. No significant correlations were found between the duration of the disease and the prevalence of any bacterial genus or species. In addition to the duration of pneumoconiosis, patients were further divided into two subgroups: working miners (*n* = 8) and former miners (*n* = 27), who stopped working due to the onset of the disease. We were also unable to find significant differences in the abundance of any genus or species of bacteria between these subgroups.

In addition, multiple linear regression (MLR) was performed to predict the relationship between the relative abundance of individual bacteria in CWP patients’ sputum and lifestyle/disease factors. As potential confounders, we considered age, current respiratory symptoms (cough, shortness of breath), living environment, current coal dust occupational exposure, alcohol use, cardiovascular disease (ischemia, arterial hypertension, etc.), bronchitis, stomach diseases (ulcer, gastritis, etc.), diabetes and obesity. In MLR using these potential confounders for patients, a significant regression was found for g. *Streptococcus* (*p*-value 0.007), *Streptococcus agalactiae* (*p*-value 0.005) with cough as the current respiratory symptom (*p*-value 0.03).

## 4. Discussion

Differences in bacterial populations in the human airways in health and disease have already been recognized as a possible contributing factor in the pathogenesis of diseases of the respiratory tract; however, the bacterial population in the airways of subjects diagnosed with coal worker’s pneumoconiosis (CWP) has not yet been investigated. Only a single recent publication reported increased levels of *Lachnospiraceae* and *Lachnoclostridium* in patients with silicosis compared to controls, but this was only demonstrated in the gut microbiome [31].

In assessing beta diversity, in the current study we observed an increase in the genus *Streptococcus* and, in particular, of the *Streptococcus agalactiae* species in individuals with CWP diagnosis (Table 3 and Table 4).

*Streptococcus agalactiae* (also known as group B *streptococcus* or GBS) is an important opportunistic bacterium that can cause pneumonia, sepsis, and meningitis in newborns and in patients with weakened immunity [32,33]. Cases of invasive GBS infections are frequently reported in the elderly and immunocompromised adults, including patients with diabetes mellitus, alcoholism, and cancer [34]. In the respiratory tract, GBS sometimes contributes to community-acquired pneumonia and empyema in adults [35]. When GBS causes pulmonary infections, it is usually defined as part of polymicrobial pneumonia [36]. GBS bacteria effectively attach to pulmonary epithelial cells and are capable of invasion. This is initiated by attachment to extracellular matrix molecules such as agglutinin, fibronectin, fibrinogen, and laminin, which facilitate their attachment to host cell surface proteins, such as integrins. Thus, the invasive potential of GBS is influenced by changes in the surface proteome of the host cells, which can be caused by various lung pathologies [37]. The molecular mechanisms of cytopathology caused by GBS bacteria in patients are currently under intensive investigation. It was shown that GBS induces the generation of reactive oxygen species (ROS) and loss of mitochondrial membrane potential [38]. In human endothelial cells, ROS species are generated via the NADPH oxidase pathway, and this is accompanied by cytoskeletal reorganization through the PI3K/Akt pathway and is generally associated with pathogen penetration, which provides evidence for the involvement of oxidative stress in *S. agalactiae* pathogenesis [39]. Separately, it is important to point out that the 97% sequence similarity threshold that we used based on the results of sequencing gives good reliability values for resolution at the genus level, but is insufficient for resolution at the species level, especially for streptococcal species that have a highly homologous 16s rRNA gene sequence. For this reason, at this stage of the study, we cannot unequivocally state that the representation of *S. agalactiae* is increased in the sputum of patients, since our 16S rRNA sequences may also include other types of streptococci. In the future, the use of quantitative PCR will allow us to answer this question.

In addition to pneumoconiosis, diseases leading to tissue fibrosis in the lungs include idiopathic fibrosis and cystic fibrosis. An increase in the abundance of *Streptococcus* in samples of the airway microbiome has been found for patients with idiopathic pulmonary fibrosis [40,41], but not for patients with cystic fibrosis [42]. Data on *Streptococcus* in the respiratory tract of COPD patients remains controversial. Different investigators report on the one hand, an increase in the representation of *Streptococcus* in the microbiota of patients with COPD compared to healthy controls [43,44] and on the other hand that no differences are observed in the abundance of this taxa with regard to this pathology [45,46,47].

Since pneumoconiosis is associated with the risk of cancer [7], it is interesting to consider the presence of *Streptococcus* in the respiratory tract of lung cancer patients. We recently reported a significant increase in the representation of *Streptococcus*, *Bacillus*, *Gemella,* and *Haemophilus* in the sputum of lung cancer patients compared to control subjects [48]. In addition to a significant increase in *Streptococcus*, we also observe an increase in the representation of the genera *Bacillus* and *Gemella* in the sputum of miners with CWP compared to control subjects (Table 3). However, this increase did not reach the threshold of significance in our study. In previous studies of patients with lung cancer, it was found that airway brushings of lung cancer patients show an increase in *Streptococcus* spp. along with a decrease in α-diversity in the affected lung microbiome, as compared with brushings from the contralateral non-cancerous lung [49]. An increase in the prevalence of *Streptococcus* was reported for sputum samples from LC patients [30,50,51], BALF samples [52], bronchoscopy [53], saliva [54], and lower airway [55]. Thus, it can be suggested that bacteria of the *Streptococcus* genus in the respiratory tract microbiome play an important role in the pathogenesis of many diseases of the bronchopulmonary system by inducing oxidative stress, inflammation, and genome instability. As follows from our data, these diseases also include pneumoconiosis induced by coal and silicate dust.

Current analysis showed no significant age-dependent differences in the representation of the *Streptococcus* taxa but that exposure to occupational hazards, such as coal dust may be the causative factor for the variation in sputum microbiomes. At this stage of the study, we did not establish a relationship between the composition of the sputum microbiome and the duration of CWP disease, and no difference in the content of certain bacterial taxa in the sputum of active and former miners was detected. It may be expected that further research based on larger study populations, will help to significantly increase the confidence levels for these factors and will contribute to a better understanding of the role of the respiratory tract microbiome in the pathogenesis of CWP.

## 5. Conclusions

This pilot study for the first time presents the results of an analysis of sputum microbiomes in a small group of current and former coal miners suffering from CWP, living in Kuzbass, a coal-mining region in Russia. The method of massive parallel sequencing of 16S rRNA genes has been used for the first time to obtain a taxonomic characteristic of the sputum samples from the coal miners with CWP diagnosis. A significant increase in *Streptococcus* spp. in the sputum of CWP patients compared to controls was found. These changes in microbial communities may contribute to the development of CWP. Based on these findings, future studies will be focused on the specific interactions between pathogen and host involved in CWP pathogenesis. In particular, digital droplet PCR will be used to test the predictability of *Streptococcus* and *Streptococcus agalactiae* (or other Streptococcal species) abundance in sputum samples from CWP patients and controls.

## Figures and Tables

**Figure 1 life-12-00830-f001:**
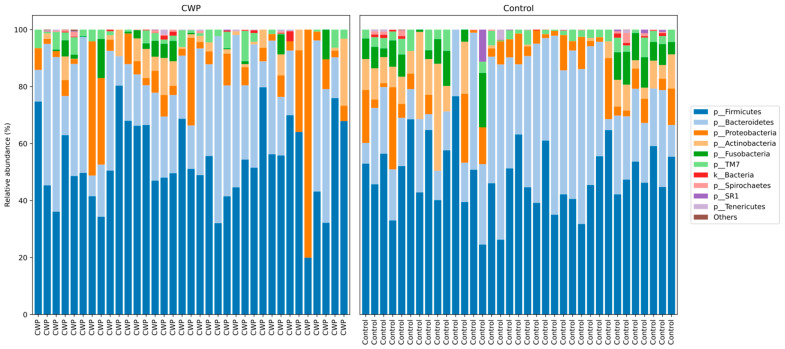
Taxonomic structure of the sputum microbiomes from CWP subjects and controls at the phyla level.

**Figure 2 life-12-00830-f002:**
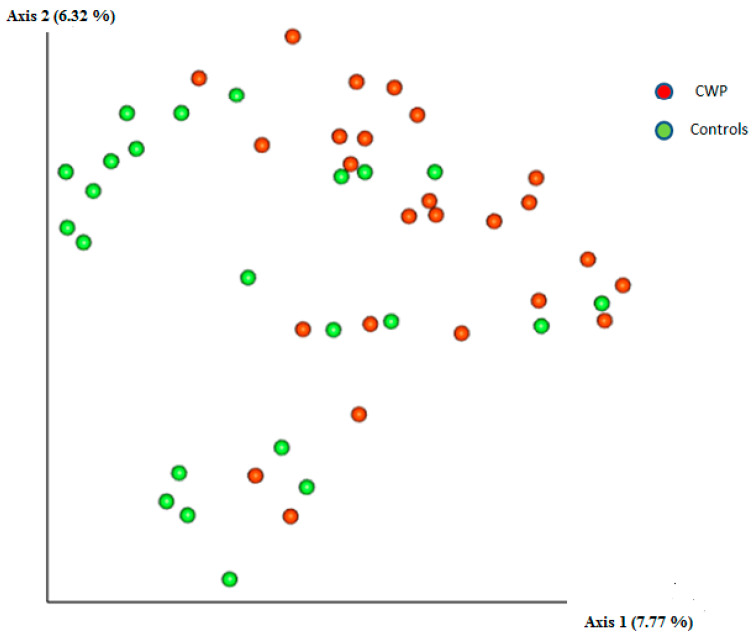
Two-dimensional diagram constructed by the method of principal components analysis demonstrating the phylogenetic similarity of prokaryotic sputum communities in CWP subjects and controls.

**Figure 3 life-12-00830-f003:**
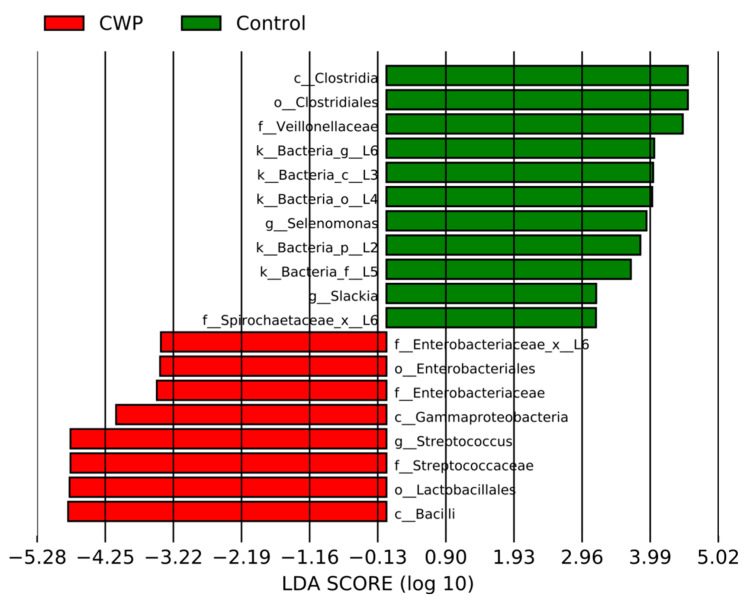
Barplots representing the effect size for particular taxa in sputum samples in the CWP and control groups. LDA-linear discriminant analysis.

**Table 1 life-12-00830-t001:** Characteristics of the study cohorts.

Variables	Coal Worker’s Pneumoconiosis, *n* = 35	Healthy Kemerovo Residents (Control), *n* = 35
Age (years) (mean ± SD)	58.5 ± 8.3	55.7 ± 11.7
Stage of CWP (%):		-
I	88.6	
II	11.4	
Respiratory symptoms (%):		
Cough	22.9	8.6
Dyspnea	5.7	5.7
Chronic diseases (%):		
Heart and vessels	34.3	34.3
Bronchitis	14.3	5.7
Stomach	14.3	11.4
Diabetes	5.7	5.7
Obesity	5.7	2.9
Living environment (%):		
City	94.3	91.4
Village	5.7	8.6
Coal dust occupational exposure (%):		
Current miners	23.0	
Former miners	77.0	-
Smoking status (%):		
Non-smokers	97.1	94.3
Smokers	2.9	5.7
Alcohol status (%):		
Non-drinker	14.3	20.0
Rare drinker	77.1	48.5
Medium drinker	8.6	31.5
Diet (%):		
Vegetarian	0	0
Non-vegetarian	100	100

**Table 2 life-12-00830-t002:** Average percentage abundance of phyla present in «core» microbiome.

Phyla	Controls, Mean ± SD	CWP, Mean ± SD	*p* Value
*Firmicutes*	47.46 ± 11.25	50.41 ± 14.1	0.39
*Bacteroidetes*	22.93 ± 10.88	19.83 ± 11.51	0.29
*Actinobacteria*	10.1 ± 7.91	8.99 ± 5.76	0.85
*Proteobacteria*	8.24 ± 6.54	11.45 ± 16.56	0.9
*Fusobacteria*	6.98 ± 4.45	4.56 ± 3.96	0.02
*TM7*	1.8 ± 1.68	1.53 ± 1.84	0.44
*Spirochaetes*	0.53 ± 0.96	0.6 ± 0.99	0.53
*Tenericutes*	0.18 ± 0.74	0.23 ± 0.41	0.1

**Table 3 life-12-00830-t003:** Average percentage abundance of genera present in «core» microbiome.

Genus	Controls	CWP	*p* Value	*p* Value (FDR)
	Mean ± SD	Mean ± SD		
*Streptococcus*	16.85 ± 11.35	25.12 ± 11.37	0.0003 *	0.0023
*Prevotella* (*f. Prevotellfceae*)	15.36 ± 7.83	13.24 ± 8.06	>0.05	0.0114
*Veillonella*	15.4 ± 11.82	12.03 ± 10.14	>0.05	0.0136
*Anaerosinus*	15.36 ± 11.86	11.56 ± 10.18	>0.05	0.0159
*Selenomonas*	4.6 ± 4.39	2.07 ± 2.1	0.02	0.0045
*Porphyromonas*	3.28 ± 3.5	3.26 ± 5.84	>0.05	0.0182
*Actinomyces*	5.95 ± 6.75	4.24 ± 3.23	>0.05	0.0205
*Megasphaera*	2.46 ± 1.43	1.45 ± 1.6	>0.05	0.0227
*Alloprevotella*	2.2 ± 2.29	2.78 ± 2.92	>0.05	0.0250
*Streptobacillus*	2.98 ± 2.99	2.3 ± 2.88	>0.05	0.0273
*Leptotrichia*	3.04 ± 3.01	2.34 ± 2.74	>0.05	0.0295
*Granulicatella*	1.01 ± 1.39	1.71 ± 1.68	0.03	0.0068
*Gemella*	2.03 ± 1.93	3.12 ± 2.19	>0.05	0.0318
*Rothia*	2.27 ± 3.02	2.65 ± 3.05	>0.05	0.0341
*Bacillus*	1.67 ± 1.7	2.67 ± 2.19	0.04	0.0091
*Atopobium*	1.41 ± 1.91	1.43 ± 1.39	>0.05	0.0364
*Pasteurellaceae*	2.34 ± 1.21	3.27 ± 2.35	>0.05	0.0386
*Fusobacterium*	1.91 ± 1.82	1.65 ± 1.42	>0.05	0.0409
*Macellibacteroides*	2.01 ± 2.58	1.37 ± 2.58	>0.05	0.0432
*Neisseria*	3.83 ± 4.91	4.72 ± 12.27	>0.05	0.0454
*Bacteroides*	1.15 ± 1.6	0.82 ± 1.78	>0.05	0.0477
*Prevotella (f. Paraprevotellacacea)*	1.17 ± 1.69	0.99 ± 1.41	>0.05	0.0500

Note: * *p* Value lesser than FDR corrected *p*.

**Table 4 life-12-00830-t004:** Average percentage abundance of species present in «core» microbiome.

Species	Controls	CWP	*p* Value	*p* Value (FDR)
	Mean ± SD	Mean ± SD		
*Streptococcus agalactiae*	16.93 ± 10.92	25.32 ± 11.55	0.0002 *	0.0016
*Anaerosinus glycerini*	14.27 ± 12.03	11.5 ± 10.22	>0.05	0.0048
*Selenomonas bovis*	4.28 ± 4.28	1.98 ± 1.99	0.03	0.0032
*Megasphaera micronuciformis*	2.35 ± 2.87	1.66 ± 1.71	>0.05	0.0065
*Prevotella histicola*	1.44 ± 1.92	2.64 ± 3.33	>0.05	0.0081
*Actinomyces hyovaginalis*	4.51 ± 5.78	2.19 ± 1.67	>0.05	0.0097
*Granulicatella balaenopterae*	1.63 ± 1.54	1.69 ± 1.66	>0.05	0.0113
*Atopobium rimae*	1.41 ± 1.91	1.44 ± 1.38	>0.05	0.0129
*Prevotella pallens*	1.54 ± 1.96	0.92 ± 1.74	>0.05	0.0145
*Rothia terrae*	1.84 ± 2.56	2.44 ± 2.97	>0.05	0.0161
*Macellibacteroides fermentans*	1.93 ± 2.43	1.38 ± 2.59	>0.05	0.0177
*Bacteroides nordii*	1.25 ± 1.58	0.75 ± 1.56	>0.05	0.0194
*Prevotella tannarae*	0.52 ± 1.08	0.7 ± 1.27	>0.05	0.0210
*Lachnoanaerobaculum orale*	0.56 ± 0.74	0.55 ± 0.63	>0.05	0.0226
*Prevotella intermedia*	0.44 ± 1.01	0.48 ± 0.7	>0.05	0.0242
*Prevotella nigrescens*	0.37 ± 0.61	0.22 ± 0.39	>0.05	0.0258
*Prevotella nanceiensis*	0.61 ± 1.09	0.43 ± 0.58	>0.05	0.0274
*Bulleidia moorei*	0.23 ± 0.3	0.29 ± 0.43	>0.05	0.0290
*Clostridium bolteae*	0.22 ± 0.5	0.32 ± 0.56	>0.05	0.0306
*Porphyromonas endodontalis*	0.67 ± 1.42	1.49 ± 5.07	>0.05	0.0323
*Vestibaculum illigatum*	0.5 ± 1.27	0.31 ± 0.65	>0.05	0.0339
*Clostridium acidurici*	0.27 ± 0.8	0.26 ± 0.73	>0.05	0.0355
*Mycoplasma zalophi*	0.3 ± 0.87	0.29 ± 0.43	>0.05	0.0371
*Moryella indoligenes*	0.27 ± 0.64	0.2 ± 0.53	>0.05	0.0387
*Peptostreptococcus anaerobius*	0.2 ± 0.54	0.23 ± 0.49	>0.05	0.0403
*Treponema amulovorum*	0.11 ± 0.24	0.07 ± 0.22	>0.05	0.0419
*Oribacterium sinus*	0.39 ± 0.61	0.13 ± 0.27	>0.05	0.0435
*Leptotrichia trevisanii*	0.1 ± 0.25	0.14 ± 0.31	> 0.05	0.0452
*Lactobacillus hamsteri*	0.09 ± 0.43	0.11 ± 0.62	>0.05	0.0468
*Bergeyella zoohelcum*	0.32 ± 0.78	0.13 ± 0.2	>0.05	0.0484
*Campylobacter rectus*	0.08 ± 0.2	0.11 ± 0.24	>0.05	0.0500

Note: * *p* Value lesser than FDR corrected *p*.

## Data Availability

The datasets obtained and analyzed during the current study are available from the corresponding author on request.

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
