# Peer review of "Sputum Microbiota in Coal Workers Diagnosed with Pneumoconiosis as Revealed by 16S rRNA Gene Sequencing"

_life, 2022, doi:10.3390/life12060830_

Round 1

Reviewer 1 Report

Dear authors,

Sputum microbiota in coal workers diagnosed with pneumoconiosis as revealed by 16S rRNA gene sequencing by Druzhinin is very interesting pilot study that identifies the microbes are responsible for lung fibrosis.  However, it needs some more additional data to strengthen the study to be publishable in LIFE.

Comment 1: The authors could present the identified microbiome nucleotide sequence will

strengthen the manuscript.

Comment 2: Please provide the % of homology of each identified species to known database as table.

Author Response

Dear Reviewer, Thank you for your Comments and Suggestions.

Comment 1: The authors could present the identified microbiome nucleotide sequence will strengthen the manuscript.

We have now included an archive with the nucleotide sequences of all samples in the supplementary materials. Please see the attachment: archive - sample-seqs.zip.

Comment 2: Please provide the % of homology of each identified species to known database as table.

Taxonomy was carried out using the GreenGenes database (version 13-8). The table (please see the attachment: archive - sample-seqs.zip; folder - tax_identity.xlsx) shows statistics on the levels of sequence identity for each taxon. The number of unique sequences is given, the minimum, maximum and the average identity among all sequences for which this taxon was determined /all sequences that were determined to belong to this taxon.

Reviewer 2 Report

I find the study interesting, and from my point of view, it would be the basis for other studies on the health of miners, especially respiratory conditions. I think the article should be accepted for publication.

Author Response

Dear Reviewer, Thank you for your attention and evaluation of our manuscript.

Reviewer 3 Report

In this paper, the authors reported that CWP group has a unique signature of microbiota compared to the control group, especially the high percentage of Streptococcus  in the CWP group, which may be involved in the pathogenesis of CWP. This funding is very interesting, since now more studies shows that microbiota are related to various types of diseases development such as cancer, autism et al. However, before its publication, there are aspects can be improved. See my concerns below:

1. About the sample collection process, how the oral microbiome contamination are avoided during the sample collection?

2. The author applied the hypervariable (V3-V4) region as the amplicon target region, did the V3-V4 is a standard region for microbiota in the sputum sample? Because some hypervariable regions are better for the characterization and some scientists suggest that sequencing multiple variable regions will provide nonbiased and comprehensive view of these complex microbiomes(PMID: 26829716).

3. What is the adjusted p-value after the FDR-correctio for the multiple comparisons? The author should add this adjusted p-value in the Table 3 and Table 4. This will be more convincing for the author’s claim that “ only differences in Streptococcus abundance reached the confidence threshold”.(line 218)

Author Response

Dear Reviewer, Thank you for your Comments and Suggestions.

Comment 1: 

About the sample collection process, how the oral microbiome contamination are avoided during the sample collection?

Although it is impossible to totally exclude the possibility of oral microbiota contamination the likelihood of such contamination can be reduced to a minimum by a thorough rinsing of the mouth before coughing up a sputum sample, as we now state in the Sample Collection section.

Comment 2: The author applied the hypervariable (V3-V4) region as the amplicon target region, did the V3-V4 is a standard region for microbiota in the sputum sample? Because some hypervariable regions are better for the characterization and some scientists suggest that sequencing multiple variable regions will provide nonbiased and comprehensive view of these complex microbiomes(PMID: 26829716).

We are aware that there is no general consensus among researchers studying the respiratory microbiome regarding the selection of the variable region of the 16S rRNA gene. However, the V3-V4 region is the most commonly used target for assessing the microbiome in the respiratory tract and in particular in sputum: Cameron et al., 2017 (10.1371/journal.pone.0177062); Bello et al., 2020 (10.1016/j.arbres.2020.05.034); Jang et al., 2021 (10.1186/s12931-021-01919-1); Huang et al., 2022 (10.1111/1759-7714.14340) et al.

Comment 3: What is the adjusted p-value after the FDR-correctio for the multiple comparisons? The author should add this adjusted p-value in the Table 3 and Table 4. This will be more convincing for the author’s claim that “ only differences in Streptococcus abundance reached the confidence threshold”.(line 218)

In tables 3 and 4, we have now added columns with p-values after the FDR-correction

Round 2

Reviewer 1 Report

Now the quality of the manuscript is improved.